# Experimental and Numerical Study of a Turbulent Air-Drying Process for an Ellipsoidal Fruit with Volume Changes

**DOI:** 10.3390/foods11131880

**Published:** 2022-06-25

**Authors:** Carlos E. Zambra, Luis Puente-Díaz, Kong Ah-Hen, Carlos Rosales, Diógenes Hernandez, Roberto Lemus-Mondaca

**Affiliations:** 1Department of Industrial Technologies, Faculty of Engineering, University of Talca, Camino Los Niches Km 1, Curico 3340000, Chile; dhernandez@utalca.cl; 2Department of Food Science and Chemical Technology, Faculty of Chemical and Pharmaceutical Sciences, University of Chile, Av. Dr. Carlos Lorca 964, Santiago 8380000, Chile; lpuente@uchile.cl; 3Institute of Food Science and Technology, Faculty of Agricultural and Food Sciences, University Austral of Chile, Av. Julio Sarrazín s/n, Valdivia 5091000, Chile; kshun@uach.cl; 4Department of Mechanical Engineering, Federico Santa Maria Technical University, Av. España 1680, Valparaíso 2340000, Chile; carlos.rosales@usm.cl

**Keywords:** convective drying process, simulation and optimization, process engineering, transport phenomena, energy consumption

## Abstract

It is common in the numerical simulations for drying of food to suppose that the food does not experience a change of volume. The few numerical studies that include volume changes assume that the shrinkage occurs symmetrically in all directions. Therefore, this effect has not been fully studied, and it is known that not considering it can be detrimental for the accuracy of these simulations. The present study aims to develop a three-dimensional model for the simulation of fruits that includes the volume changes but also takes into consideration the asymmetry of the shrinkage. *Physalis peruviana* is taken as the subject of study to conduct experiments and imaging analyses that provided data about the drying kinetics and asymmetric shrinkage mode. The effective diffusion coefficient is found to be between 10^−12^ m^2^ s^−1^ and 1.75 × 10^−9^ m^2^ s^−1^. The shrinkage occurs essentially in only one direction, with an average velocity of 8.3 × 10^−5^ m/min. A numerical modelling scheme is developed that allows including the shrinkage effect in computer simulations. The performance of the model is evaluated by comparison with experimental data, showing that the proposed model decreases more than 4 times the relative error with respect to simulations that do not include volume changes. The proposed model proves to be a useful method that can contribute to more accurate modeling of drying processes.

## 1. Introduction

Drying is a very common preservation method used to obtain high-quality and stability in the final products, which is strongly dependent on process variables [1]. The convective drying is the most common and studied process and the simplest one to operate [2]. An appropriate prediction of the drying process can be an important tool for minimizing problems such as damage to the product, excessive consumption of energy, excessive wear of the drying equipment, and decrease in yields [3].

Usually, fruit is considered as a porous medium [4]. Existing mathematical models can be classified into single-phase models and multiphase models. In multiphase models, water, steam and air are considered to coexist in the pores. Recently this type of model has been used to study the convective drying of apple slices [5] and sweet potato [6]. These models, which reproduce in detail the thermodynamic process of water loss from food, and which are theoretically more precise than one-phase models, require large computational times. Therefore, they are generally solved in only one or two dimensions [4,5,6]. Such models consider the water transfer properties in the food as a function of thermodynamic variables (e.g., capillary diffusivity of water as a function of dry basis moisture content or the effective diffusivity of gas as a function of the gas saturation), and must be extended to include particular complexities of each food such as anisotropy, different pore sizes, distribution of seeds and kernels, and resistance to mass transfer through the shell or chemical reactions due to the process, etc.

On the other hand, the one-phase models simplify the process considering that the pores of the food contain only water, which is transported to the surface where it is evaporated and then carried away by the air. In the one-phase models, an effective diffusion coefficient is applied for the transport of water in the food, which is commonly obtained from experimental data. This effective diffusion coefficient allows for the particularities of the food and the dryer, and therefore very good approximations have been obtained when using them in two-dimensional (2D) [7] or three-dimensional (3D) geometries [8,9,10]. Since a reduced set of equations must be solved compared with the multiphase model, the computational times of one-phase models are always lower.

To capture appropriately the physics of the process, a coupled mathematical model of heat and mass transfer both within the solid food and surrounding airflow should be considered. Moreover, taking also into consideration some physical change, such as the 3D food shrinkage, would allow an appreciable improvement in the accuracy of computations, since that is physically more realistic.

Shrinkage during drying does not only occur in food, and it has been experimentally studied with a multiphase model and 2D computational simulation for clays using the finite volume method [11]. However, food is more complex than just porous materials. A very simple model was used to analyze the shrinkage phenomenon during the drying of sweet cherry fruits [12]. Recently, several articles on mathematical modelling including the simulations of food shrinkage in 1D and 2D with commercial software have been published [4,5,7,13]. These studies include porosity models for food, as well as one-phase and multiphase models.

We are not aware of any studies that consider the direction of the shrinkage in the context of a 3D simulation of drying. Therefore, there are still gaps in our knowledge about this aspect in the drying of food. Further study on this subject contributes to the improvement of the simulations and thus to better predictions for the drying process of different foods. In this work, the drying of fruits with three-dimensional asymmetric shrinkage is studied numerically.

Many fruits such as berries, apples, peaches, walnuts, hazelnuts, grapes, etc., are roughly in the shape of an ellipsoid. In this paper the *P. peruviana* is taken as a study case. This fruit is native to the Andean highlands and belongs to the *Solanaceae* family [2]. It contains seeds distributed inside and its geometry approximates an ellipsoid. In this work, a coupled model of turbulent convection of heat and mass is implemented for the airflow around an ellipsoidal fruit. There is diffusion of heat and mass within this fruit, which is considered as a porous medium of one phase that loses volume along a preferential direction, which was determined after analyzing the results of an imaging procedure developed for this study.

Since drying processes present high energy consumptions, a proper selection of operational parameters is important for low environmental impact and sustainability of them. In this context, energy consumption has been studied in food drying processes, such as for cherry fruits [14], berberis fruits [15], pomegranate [16] and squash seeds [17] in order to respond to the challenge of reducing the energy demand of the agroindustry. Hence, in this work we study additionally some energetic aspects considering different drying conditions and energy input from the dryer.

## 2. Materials and Methods

This section presents first the characterization of the fruit and the experimental methodology used to obtain the drying curves, volume change and shrinkage images. Then, the mathematical model is explained with its respective initial and boundary conditions. Finally, the numerical and grid selection procedure are presented.

### 2.1. Raw Material

The ellipsoidal fruit used for the drying experiments was *P. peruviana*. Extensive physical-chemical characterizations for the *P. peruviana* can be obtained from several references [2,18,19,20]. These fruits were purchased in the town of Olmué, Valparaíso Region, Chile. The fruits used were selected to provide a homogeneous group based on the date of harvest, color, size, and freshness according to visual analysis. Then, they were kept refrigerated at 5 °C by one week until the drying process. The fruits had a major diameter of 38 mm and a minor diameter of 36 mm. The moisture content was determined by AOAC method no. 934.06 [21], employing a vacuum oven (OVL570, Gallenkamp, Loughborough, UK) and an analytical balance accurate to ±0.0001 g (Jex120, CHYO, Tokyo, Japan).

### 2.2. Drying Experiments

Figure 1a shows the physical set-up used for the experiments and the numerical simulations. The experiment was performed in a tunnel dryer at four different drying air temperatures: 328 K (55 °C), 338 K (65 °C), 348 K (75 °C) and 358 K (85 °C), at a constant inlet velocity of (1.0 ± 0.1) m/s. Fruit samples were placed in a stainless-steel basket hanging from a balance with a ±0.01 g accuracy (SP402 Scout-Pro, Ohaus, Shanghai, China). The balance was communicated with an interface system (SP232 Scout-Pro, Ohaus, Parsippany, USA), where the weight changes in real time were recorded and stored by Microsoft HyperTerminal 6.2 software for Windows 7 (Redmond, Washington, USA). The weight of fruit was recorded at 15 min intervals during the first four hours of drying and then every 1 h until reaching a constant weight (equilibrium condition).

The dimensions of the experimental dryer are 0.25 m × 0.25 m × 0.25 m. The walls are adiabatic and impermeable. The air inlet section (measuring 0.036 m × 0.036 m) is located at the center of one of the walls, while the air outlet is located at the center of the opposite wall and has the same dimensions as the inlet.

We use Cartesian coordinates (*x*, *y*, *z*) for reference in the simulations. The *x*-axis is aligned with the direction of the inlet flow (i.e., is the streamwise direction), *y* is the vertical direction and *z* is the spanwise direction. Hence, the inlet is located at coordinates (*x* = 0, *y* = 0.107 to 0.143, *z* = 0.107 to 0.143) and the outlet at (*x* = 0.25, *y* = 0.107 to 0.143, *z* = 0.107 to 0.143). As mentioned before, the shape of the fruit can be approximated by an ellipsoid whose main semi-axes measured 0.018 m, 0.019 m and 0.019 m at the initial time. These values agree with those reported in the reference [20].

The shortest main axis was aligned with the *x*-direction, and the center of the ellipsoid was located at (*x_c_* = 0.125, *y_c_* = 0.125, *z_c_* = 0.125).

### 2.3. Volume Determination

Volume change of the fruit during the drying process was determined by the solid displacement method using poppy seeds [22]. We selected this method because it is one of the most recommended to avoid contaminating the fruit. The seeds weight (displaced solid) that uniformly leveled a glass container of known volume (*Vol_c_*_,1_) and weight (*m_c_*_,4_) was determined. The *P. peruviana* sample, previously weighed (*m_f_*_,3_), was placed in the container together with the seeds, and both were weighed (*m_sample_*_,2_). The seeds weight (*m_s,s_*_,1_) in the container was obtained with the following equation [23]:(1)$$m_{s,s,1}=m_{sample,2}-m_{f,3}-m_{c,4}$$

The seeds volume (*V_s,s_*_,3_) was obtained by dividing the seeds weight by its density (1100 kg/m^3^). Finally, the experimental volume of the fruit at time *t* (*Vol_f,exp_*(*t*)) was determined by the following equation:(2)$$Vol_{f,exp}\left(t\right)=Vol_{s,1}-Vol_{s,s,3}$$

### 2.4. Image Capture

An image capture mechanism was built with a lighting system consisting of two fluorescent tubes of 6 watts and 6500 K color, and a rotating stand to take pictures of the *P. peruviana* samples from the front and both side angles. The darkroom was accommodated as reported in the reference [24]. A Canon^®^ Power Shot A520 digital camera (at 1600 × 1200 pixels resolution) was placed at 5 cm in front of the fruit. The images were taken every 5 min at an air inlet temperature of 338 K (65 °C). Using a subroutine in MATLAB^®^ v.6.0, the images obtained in the darkroom were processed so that only the fruit retained its own color while the rest of the image was assigned a black value.

### 2.5. Mathematical Modeling

The mathematical model applied in this work includes a 3D unsteady transport of momentum, thermal energy and moisture concentration in the drying air, which is solved simultaneously with the 3D diffusion of heat and moisture content in the shrinking fruit. The meaning of the variables in the following equations and its units can be consulted in Nomenclature.

#### 2.5.1. Mathematical Model for Airflow in the Dryer

For typical conditions in the dryer the flow is turbulent. Given the small area of the inlet compared with the dryer, the flow enters as a jet. For the inlet velocity of 1 m/s and the inlet hydraulic diameter of 0.036 m, a jet Reynolds number of the order of 2000 results. It is known [25,26] that at this Reynolds number a jet is unstable and transition to turbulence will occur. In our case, the flow is also perturbed by the impingement on the fruit and the deviation towards the equally small outlet, so that a relaminarization does not occur in the available space.

The effects of turbulence are included by the usual decomposition of each transported variable into the sum of a statistical mean value plus turbulent fluctuations, followed by the averaging of the corresponding transport equation. The airflow is treated with variable density but quasi-incompressible (i.e., low-Mach number approximation), so that density changes are associated with temperature variations, but they are not directly coupled with pressure. In this way, the governing equations for the average air flow [8,9,27] include the mass conservation (or continuity) equation
(3)$$\frac{\partial \bar{\rho }}{\partial t}+\nabla \cdot \left(\bar{\rho }\;\bar{u}\right)=0$$
and the conservation equation for linear momentum
(4)$$\frac{\partial \left(\bar{\rho }\;\bar{u}\right)}{\partial t}+\nabla \cdot \left(\bar{\rho }\;\bar{u}\;\bar{u}\right)=-\nabla p+\nabla \cdot \left[\left(\mu_{a}+\mu_{t}\right)\left(\nabla \bar{u}+\nabla {\bar{u}}^{T}\right)\right]+\bar{\rho }g$$
where the overbars denote mean values, **u** is the flow velocity vector field, *p* is the pressure, **g** is the gravitational acceleration, *μ_a_* is the dynamic viscosity of air and *μ_t_* is the local turbulent viscosity. This variable is introduced as a closure for the turbulent stresses arising from the velocity decomposition. The isotropic part of the turbulent stress tensor has been absorbed into the pressure term.

These Equations (3) and (4), are coupled with the transport equations for thermal energy and moisture content in the air. The energy equation, expressed for the temperature as the dependent variable, is written as
(5)$$\frac{\partial \left(\bar{\rho }\;\bar{T}\right)}{\partial t}+\nabla \cdot \left(\bar{\rho }\;\bar{u}\;\bar{T}\right)=\nabla \cdot \left[\left(\bar{\rho }\alpha_{a}+\frac{\mu_{t}}{{\mathrm{Pr}}_{t}}\right)\nabla \bar{T}\right]$$
where *α_a_* is the thermal diffusivity of air and Pr*_t_* is a turbulent Prandtl number applied as a closure for the turbulent heat diffusion. Similarly, for the transport equation of moisture content in the air we have
(6)$$\frac{\partial \left(\bar{\rho }\;\bar{C}\right)}{\partial t}+\nabla \cdot \left(\bar{\rho }\;\bar{u}\;\bar{C}\right)=\nabla \cdot \left[\left(\bar{\rho }D_{a}+\frac{\mu_{t}}{{\mathrm{Sc}}_{t}}\right)\nabla \bar{C}\right]$$
where *D_a_* is the molecular mass diffusivity of moisture in air and Sc*_t_* is a turbulent Schmidt to account for the turbulent mass diffusion.

To compute the turbulent viscosity, we apply the standard *k*–*ε* turbulence model, which consists of a transport equation for the turbulent kinetic energy *κ*,
(7)$$\frac{\partial \left(\bar{\rho }\kappa \right)}{\partial t}+\nabla \cdot \left(\bar{\rho }\;\bar{u}\;\kappa \right)=\nabla \cdot \left[\left(\mu_{a}+\frac{\mu_{t}}{\sigma_{\kappa }}\right)\nabla \kappa \right]+G-\bar{\rho }\varepsilon$$
and a transport equation for the turbulent kinetic energy dissipation rate *ε*,
(8)$$\frac{\partial \left(\bar{\rho }\varepsilon \right)}{\partial t}+\nabla \cdot \left(\bar{\rho }\;\bar{u}\;\varepsilon \right)=\nabla \cdot \left[\left(\mu_{a}+\frac{\mu_{t}}{\sigma_{\varepsilon }}\right)\nabla \kappa \right]+GC_{1}\left(\frac{\varepsilon }{\kappa }\right)-\bar{\rho }C_{2}\left(\frac{\varepsilon^{2}}{\kappa }\right)$$

*G* represents the production rate of turbulent kinetic energy (W m^−3^) by the action of the mean-flow velocity gradient tensor against the turbulent stresses, and in the quasi-incompressible approximation is given by
(9)$$G=\left(\mu_{a}+\mu_{t}\right)\left(\nabla \bar{u}+\nabla {\bar{u}}^{T}\right)\;:\nabla \bar{u}$$

The turbulent viscosity is obtained then as
(10)$$\mu_{t}=\rho C_{u}\left(\frac{\kappa^{2}}{\varepsilon }\right)$$

Parameters *C*_1_, *C*_2_, and $C_{u}$ are closure constants of the model, as well as *σ_κ_* and *σ_ε_* which are equivalent to the turbulent Prandtl numbers for the turbulent diffusion of kinetic energy and its dissipation rate. The values for the standard *k*–*ε* model are [28]: (11)$$C_{1}=1.44;\;C_{2}=1.92;\;C_{u}=0.09;\;\sigma_{\kappa }=1.00;\;\sigma_{\varepsilon }=1.217;$$

The turbulent Prandtl and Schmidt numbers are known to be of order one [29,30]. These parameters measure the ratio between the turbulent viscosity and the turbulent diffusivities for heat or mass respectively. In the present study, the flow is mostly of the free shear kind, for which appropriate values are [31,32,33].
(12)$$Pr_{t}\approx 0.5\;\;;\;Sc_{t}\approx 1.0$$

The air thermophysical properties are calculated as functions of temperature [8] as
(13)$$\nu_{a}=\left(-1.155\times 10^{-14}\right)T^{3}+\left(9.572\times 10^{-11}\right)T^{2}+\left(3.760\times 10^{-8}\right)T-\left(3.448\times 10^{-6}\right)\;{\left[m^{2}{\;s}^{-1}\right]}^{\;}$$
(14)$$\alpha_{a}=\left(9.10\times 10^{-11}\right)T^{2}+\left(8.82\times 10^{-8}\right)T-\left(1.07\times 10^{-5}\right)\left[m^{2}{\;s}^{-1}\right]$$
(15)$$D_{a}=2.50\times 10^{-5}\left[m^{2}{\;s}^{-1}\right]$$

#### 2.5.2. Mathematical Model for Fruit in the Dryer

We treat the transport phenomena within the fruit as a conjugate problem with the processes taking place in the drying air. Within the fruit, diffusion of heat and moisture occur simultaneously. Therefore, the model includes an equation for the 3D unsteady heat diffusion in the ellipsoidal fruit (*P. peruviana*),
(16)$${\left(\rho C_{p}\right)}_{f}\frac{\partial T}{\partial t}=k_{f}\;\nabla^{2}T$$
coupled with the 3D unsteady equation for moisture concentration in the fruit,
(17)$$\frac{\partial C}{\partial t}=D_{f}\left(T_{in},{\bar{C}}_{w}\right)\;\nabla^{2}C$$

The *ρ*, *C_p_*, and *k* values of this fresh fruit at ambient temperature are 997.3 kg m^−3^, 1666 J kg^−1^ °C^−1^ and 0.5 W m^−1^ °C^−1^, respectively. These values were taken from references [18,20] where the fruits had been harvested and stored in the same conditions that we had for the raw material of our study. The moisture diffusivity *D_f_* (*T_in_*, ${\bar{C}}_{w}$) was determined experimentally and presents a strong dependence on the inlet temperature of the air to the dryer, *T_in_*. For the numerical model it is calculated by an Arrhenius-type relation
(18)$$D_{f}\left(T_{in},{\bar{C}}_{w}\right)=A\left({\bar{C}}_{w}\right)\cdot e^{\left(\frac{-E_{a}}{RT_{in}}\right)}$$
where *R* is the universal gas constant (8.314 × 10^−3^ kJ mol^−1^ K^−1^), *E_a_* is the activation energy (25,400 kJ mol^−1^), and the pre-exponential factor $A\left({\bar{C}}_{w}\right)$ (m^2^ s^−1^) is a function of the mean moisture concentration in the fruit $\left({\bar{C}}_{w}\right)$. This factor can be obtained from the experimental drying curves, as explained later in Section 3.2.2.

#### 2.5.3. Initial and Boundary Conditions

The initial air velocity inside the dryer was assumed to be zero. The values for the air inlet velocity in the *x* coordinate and air moisture content were *V_in_* = 1 m s^−1^ and *C_in_* = 0.011 (w.b.) and correspond to the equilibrium moisture reached between air and fruit. These parameters were kept constant during the simulations. Four prescribed temperatures of the air at the inlet (*T_in_*) were considered: 328 K (55 °C), 338 K (65 °C), 348 K (75 °C) and 358 K (85 °C), which match the temperatures used in the experiments. An initial constant temperature of *T_a_* (*x*, *y*, *z*, 0) = 303 K of the air inside the dryer was assumed. At the outlet, zero-gradient outflow boundary conditions were imposed (*∂*(⋅)/*∂x* = 0). The walls were considered adiabatic and impermeable. The initialization of the *κ* and *ε* fields was based on the initial velocity at the inlet of the dryer and the length scale of the dryer (*L* = 0.25 m).
(19)$$\kappa \left(x,y,z,0\right)=0.01{\left(V_{in}\right)}^{2}$$
(20)$$\varepsilon \left(x,y,z,0\right)=\frac{{\left(V_{in}\right)}^{3}}{0.2L}$$

For the cells located inside the fruit at any time, the velocity field is zero. An initial homogeneous temperature of 293 K (20 °C) was considered for the fruit for all the cases studied. The initial moisture content of the fruit is ${\bar{C}}_{w,\;initial}$ = 0.8051 (w.b.) which corresponds to the initial moisture measured experimentally.

Classical wall functions for the standard *κ*–*ε* turbulence model were applied to the flow conditions in the cells adjacent to the walls or the solid fruit [8]. These functions are based on the law-of-the-wall region of the boundary layers, whereas the *κ*–*ε* model takes control further from the surface and in the free stream region. A complete explanation of the wall functions for velocity and temperature used in our computer code may be found in the reference [27].

### 2.6. Numerical Procedure and Grid Selection

The mathematical model based on the coupled system of non-linear partial differential equations described previously is solved using the finite volume method [34,35]. The development of the numerical simulation is based on a non-commercial in-house computer program written in Fortran, which has been fully validated in previous published works (e.g., Refs. [8,9] and references therein). The governing equations were treated in the generalized form for the transport of a variable φ, with unsteady convection, diffusion, and source terms (Equations (3)–(8), (16), and (17)):(21)$$\frac{\partial \left(\bar{\rho }\varphi \right)}{\partial t}+\nabla \cdot \left(\bar{\rho }\;\bar{u}\;\varphi \right)=\nabla \cdot \left(\Gamma \;\nabla \varphi \right)+S_{c}+S_{p}\varphi$$
which are integrated in the finite volume cells. The processes in the air flow and within the fruit are treated as a conjugate problem. Thus, in the cells occupied by the fruit the velocity $\bar{u}$ vanishes and heat and mass diffusion become the only transport mechanisms. The time integration is performed with an implicit Euler scheme. For each time step (0.01 s in this work), the governing equations are coupled sequentially (external iterations) with the SIMPLE algorithm [27]. Under-relaxation coefficients are applied to these external iterations, with a value of 0.5 for all the variables, except for the pressure correction, for which a value of 0.9 was used instead. The discretized equations for each dependent variable φ are solved iteratively (internal iterations) with a line Gauss-Seidel method combined with a Tridiagonal Matrix Algorithm (TDMA). The convergence criterion for these internal iterations was a maximum difference of 1 × 10^−4^ between successive solutions at any control volume, for all the variables.

The grid is selected on the basis of previous studies of drying of the olive waste cake [8] carried out in a dryer twice the size (0.5^3^ m^3^) of the one utilized in the current research. In that research [8], an exhaustive grid convergence study, using a mathematical model similar to the one used in the present study (but without shrinkage), was made. It was found that a non-uniform grid of 70 × 80 × 70 cells and a time step of 0.01 s are appropriate to assure the convergence of the numerical algorithm. Based on this previous study, three non-uniform meshes of 72 × 62 × 62 cells (grid A), 80 × 70 × 70 cells (grid B), and 95 × 83 × 83 cells (grid C) in *x*, *y*, *z* coordinates, and three-time step 0.01 s, 0.05 s and 0.1 s, are analyzed for this work. The grid is finer in the region occupied by the fruit and in its neighborhood. For the grid convergence analysis we do not apply a shrinkage model for the fruit, and a constant temperature and moisture content are prescribed for it. It was considered that the steady state had been reached when the maximum variation, for the three velocity components (*u*, *v*, *w*), remained lower than 1 × 10^−4^ during 50-time steps. Since the greatest velocity gradients occur in the *x*-direction, the grid convergence assessment is done by comparing the distribution of the *u* velocity component along a line directed in the *x*-direction (at *z* = 0.125 m; *y* = 0.125 m). Taking the finest grid as reference, the relative error (RE) in the *u* velocity distribution with respect to grid C was calculated for grids A and B. RE is computed as $\int_{0}^{xmax}\left|u\left(x\right)-u_{C}\left(x\right)\right|dx/\int_{0}^{xmax}\left|u_{C}\left(x\right)\right|dx$, where *u_C_*(*x*) is the reference velocity distribution. The integrals are calculated by the trapezoidal method.

The results were RE = 0.2 and RE = 0.03 for grids A and B respectively for a time step of 0.01 s. The RE increases as the time step increases. Consequently, a non-uniform grid of 80 × 70 × 70 cells (grid B) with a time step of 0.01 s was selected and used in all the simulations presented in this work. This grid is shown in Figure 2b and it provides 4500 cells inside the initial volume of the fruit.

### 2.7. Energetic Analysis

As mentioned in the Introduction, drying processes are characterized by high specific energy consumptions. Hence, it is of interest to apply our results to assess the energy consumption of *P. peruviana*. The values for moisture content and drying time come from data generated by the computer simulations.

#### Energy Consumption and Efficiency

The energy consumption (*E_T_*) represents the amount of energy, per unit mass, consumed specifically for the drying of fruit [16]. It can be calculated as
(22)$$E_{T}=\frac{V_{a}\;\rho_{a}\;A\;Cp_{a}\;\Delta T\;t}{m_{0}}$$
where *V_a_* is the air inlet velocity (m s^−1^), *ρ*_a_ the air inlet density (kg m^−3^), *A* the inlet area (m^2^), *t* the total drying time (s), Δ*T* the temperature difference (°C) between air inlet and outlet, *Cp_a_* the specific heat of the air (kJ kg^−1^ °C^−1^) at constant pressure and *m*_0_ the initial mass of the fruit. We remark that the definition (22) only considers the heat transferred to the fruit but does include the additional energy required for heating the drying air. The purpose is to obtain the specific energy consumption per unit mass of fruit, independent of the particular dryer and dependent only on the type of fruit. This makes easier the comparison of the specific heat of drying between different materials.

In addition, energy efficiency (*η_E_*) is defined as the ratio of the energy required for evaporation of moisture from the food to the total energy consumed, that is
(23)$$\eta_{E}=\frac{Q_{w}}{E}=\frac{h_{fg}\;m_{w}}{E}$$

Here, *h_fg_* is the latent heat of vaporization and *m_w_* the mass of evaporated water. This mass is calculated as
(24)$$m_{w}=\frac{m_{0}\left(x_{o}-x_{wf}\right)}{1-x_{wf}}$$
where *x*_0_ and *x_wf_* are the initial and final moisture content in wet basis (kg_water_ kg^−1^ w.b.), respectively. For the latent heat of vaporization (J/kg) we apply the following expression
(25)$$h_{fg}=\left\{\begin{array}{c} 2.503\times 10^{6}-2.386\times 10^{3}\left(T-273.16\right)\;273.16\;K\leq T\leq 338.72\;K\\ {\left[7.33\times 10^{12}-1.60\times 10^{7}T^{2}\right]}^{0.5}\;\;\;338.72\;K\leq T\leq 533.16\;K \end{array}\right.$$
where *T* is the temperature of the drying air.

## 3. Results and Discussions

This section presents and discusses first the experimental results for the characterization of *P. peruviana*, volume changes, drying curves, and image analysis. Based on these results, an experimental-numerical methodology is proposed to reproduce the experimental results of weight loss (drying curves) and volume change (shrinkage). Then, the numerical results of drying *P. peruviana* are presented and discussed.

### 3.1. Experimental Results

#### 3.1.1. Fruit Characterization and Volume Changes

The initial moisture content of the fruit (after storage) was (80.51 ± 0.10)% (w.b.). This moisture agrees with values reported in previous studies for *P. peruviana* [2,20]. The moisture measurements during the drying were made for each air inlet temperature (328–358 K) until reaching an equilibrium moisture content of (11.1 ± 0.30)% (w.b.) for each air inlet temperature.

To characterize the change of volume of the fruit as the moisture content is varying, we plot in Figure 2a the experimental values of the volume ratio ($Vol_{f,exp}/Vol_{f,initial}$) as a function of the instantaneous moisture content ratio ($C_{w}\left(t\right)/C_{w,initial}$). The scattering of the data is small, and the differences among the volume ratios for different temperatures are within a range of ±0.02 when evaluated at the same moisture ratio. A linear trend relation between these dimensionless ratios can be observed in the plotted data. This behavior has also been documented by other authors for diverse spherical foods [12,13,36]. However, the range for the volume ratio differs significantly of other reported fruits. In reference [12], volume variations between 1 and 0.24 are reported for cherries, whereas the volume rate found in this work for *P. peruviana* varies between 1 to 0.81. Interestingly, both cherries and *P. peruviana* have similar geometry and size. This difference in the range of the volume rate might be associated with a different structure and composition of both fruits. Cherries have a homogeneous pulp with a pit in the center, which does not initially intervene in the volume change process, while the *P. peruviana* contains seeds distributed throughout the pulp, which limit the volume change of the fruit.

The first two columns in Figure 3 show front and lateral image captures of the fruit sample during the drying process with air at an inlet temperature of 338 K (65 °C), taken at *t* = 0, 90, 180, 360, and 720 min. In the images of the frontal view it is observed that there is no shrinkage in the spanwise (*y*,*z*) plane and there is only a slight change in color and roughness of the fruit surface. The lateral view images present the (*x*,*y*) plane, showing clearly that there is shrinkage along the *x* coordinate. During the drying process, the seeds were displaced toward the central middle zone of the dried fruit, which considerably influenced the final shape of the dehydrated fruit. In these images, (*x*,*y*) are local coordinates with the origin at the center of the fruit. To measure the shrinkage, straight lines are drawn along the *y*-direction so that they touch the surface of the fruit at *y* = 0 on each side of the fruit. These intersections determine the length of the minor semiaxis of the ellipsoid. The values of the semiaxis obtained in this way on both sides were averaged, resulting in 0.0141 m, 0.0076 m, 0.0046 m and 0.0045 m for 90, 180, 360 and 720 min respectively. The behavior corresponds to an average shrinkage velocity of 8.3 × 10^−5^ m/min or 8.3 × 10^−3^ cm/min.

Therefore, the experimental data obtained by the image capture provides very useful information about the highly non-isotropic volume change of the *P. peruviana* fruit when subjected to convective drying. This knowledge is essential for the development of a more accurate model for the computational simulation of the drying process of this kind of fruit. The relevant data that will be used for the numerical solution are its rate of shrinkage and the preferred direction in which it occurs.

#### 3.1.2. Drying Rates

Figure 2b shows the experimental variation of the moisture ratio with time, together with the same ratio calculated using computational solutions with and without shrinkage of the fruit. These last results will be discussed later in Section 3.2.3. The experimental times to reach equilibrium moisture content were 900, 780, 480, and 300 min at the air inlet temperatures of 328, 338, 348, and 358 K, respectively (55, 65, 75 and 85 °C). The observations show the strong effect that the temperature has on the drying process, and as expected, as the drying air temperature was increased, the drying rate increased and the drying time decreased. The shape of these curves are similar to those found for others berries such as strawberries [37], blueberries [38], and sweet cherry [39]. The drying curves and drying times obtained are close to those presented in previous works for drying of *P. peruviana* with an air inlet temperature from 333 K to 363 K [40].

### 3.2. Numerical Simulation

Having studied experimentally the drying of *P. peruviana* fruit, we develop in the following a computational procedure for the non-isotropic shrinkage process, which is based on the mathematical model presented in Section 2.4 and incorporates the particular behavior observed experimentally for this fruit.

#### 3.2.1. Shrinkage Procedure for the Fruit

The experiments clearly show that the fruit undergoes a significant shrinkage during drying. Moreover, this shrinkage occurs mainly in one direction, which is aligned with the streamwise direction of the drying air (see Figure 3).

In order to include these effects in the numerical model, we postulate a linear correlation function between the instantaneous volume ratio and its corresponding moisture content ratio:(26)$$\frac{Vol_{f,exp}\left(t\right)}{Vol_{f,initial}}=Coef1\left(\frac{{\bar{C}}_{w}\left(t\right)}{{\bar{C}}_{w,initial}}\right)+Coef2$$

This is based on the experimental trend observed in Figure 2a. The coefficients *Coef*1 and *Coef*2 of this equation were obtained by linear regression for each air inlet temperature using the experimental data of volume ratio and moisture concentration ratio. This procedure gave us four similar functions. A mean value of the moisture content ratio was calculated from the four experimental moisture content ratios at each time *t* (this *t* time is the sampling time). These mean values were compared with the estimation given by each regression function. The function with the lowest error (R^2^) was chosen to represent the volume change in simulations. In this way, the resultant coefficients are
(27)Coef1=0.1954 ; Coef2=0.8062
which correspond to the case with air inlet temperature at 338 K (R^2^ = 0.9923). The equation and the respective curve with the best fit are shown in Figure 2a.

With regard to the shrinkage, the procedure we apply to reproduce this effect in the simulations is the following. First, the initial 3D region occupied by the fruit in the computational domain is determined by the equation of an ellipsoid centered at position (*x_c_*, *y_c_*, *z_c_*),
(28)$$\frac{{\left(x-x_{c}\right)}^{2}}{a_{i}{}^{2}}+\frac{{\left(y-y_{c}\right)}^{2}}{b_{i}{}^{2}}+\frac{{\left(z-z_{c}\right)}^{2}}{c_{i}{}^{2}}=1$$
where *a_i_*, *b_i_* and *c_i_* are the semi-axes. All the cells inside the region defined by Equation (29) are initially marked as belonging to the fruit. For this particular case, the initial values for the *P. peruviana* fruit are *a_i_* = 0.018 m, *b_i_* = 0.019 m and *c_i_* = 0.019 m. The volume corresponding to this ellipsoid is
(29)$$Vol_{w,eff}\left(t\right)=a_{i}\;b_{i}\;c_{i}\left(\frac{4\pi }{3}\right)$$
which depends on time *t* essentially because the minor semi-axis *a_i_*, aligned with the streamwise direction of the external flow, is a function of time and will contract during the drying, as observed in the experiments. Then, at each time step in the simulations the volume calculated by Equation (30) is compared with the instantaneous volume calculated by the experimental correlation (27). If the difference exceeds a given tolerance, then the semi-axis *a_i_* is reduced by the minimal length decrement resolvable in the grid (which corresponds to the spatial discretization Δ*x* in that region of the domain). More specifically, the criterion for applying this reduction in the semi-axis is when
(30)$$\frac{Vol_{w,eff}\left(t\right)-Vol_{f,exp}\left(t\right)}{Vol_{f,initial}}\geq 0.02$$

The tolerance of 0.02 comes from the amplitude of the scattering of experimental data for the ratio ($Vol_{f,exp}/Vol_{f,initial}$) noticed in Section 3.1.1. In this way, the volume and shape of the fruit in the simulations conform, in a discretized mode, to the experimental behavior. The numerical shrinkage rate ranged between 5.5 × 10^−5^ m/min and 9 × 10^−5^ m/min, which are consistent with the order of the experimental value.

We remark that this procedure could be used for any shrinking fruit, once the corresponding functions have been determined.

#### 3.2.2. Pre-Exponential Factor and Calculation Algorithm

The pre-exponential factor $A\left({\bar{C}}_{w}\right)$, used to calculate the effective diffusion coefficient (Equation (18)), for the moisture diffusivity, is determined from the experimental data, for each inlet temperature of air. The procedure is summarized in the flowchart shown in Figure 4. At each time step during a computation that reproduces a drying curve, the mean moisture content in the fruit, determined by the finite volume method, is compared with the corresponding experimental value, and the factor $A\left({\bar{C}}_{w}\right)$ is iteratively adjusted until there is agreement (within a prescribed precision) between the computed and experimental values at that instantaneous state. This procedure gives a dataset of pre-exponential factors as a function of ${\bar{C}}_{w}$ which can fitted by a sixth-degree polynomial:(31)$$A\left({\bar{C}}_{w}\right)=a+b{\bar{C}}_{w}^{\;}+c{\bar{C}}_{w}^{\;2}+d{\bar{C}}_{w}^{\;3}+e{\bar{C}}_{w}^{\;4}+f{\bar{C}}_{w}^{\;5}+g{\bar{C}}_{w}^{\;6}$$

The coefficients obtained for this equation, for all the cases, are presented in Table 1. These functions for $A\left({\bar{C}}_{w}\right)$ can be used in simulations with and without shrinkage.

#### 3.2.3. Comparison between Experimental and Calculated Drying Curves

The Figure 2b shows the computed variation of the moisture ratio with time, together with the experimental measurements discussed previously. The computations are performed with two methods: the traditional mathematical model (Calc. Tr.), which does not consider the shrinkage of the fruit, and the model introduced in this work that includes the shrinkage effect (Calc.).

The mathematical model without shrinkage gives a slower decrease in moisture content compared with the experiments. On the other hand, the method proposed in this work, endowed with a shrinkage model, can predict accurately the moisture reduction rates. The relative error between the computed and experimental curves is calculated in a way analogous to the description in Section 2.6, as the integral of the absolute value of the difference between computed and experimental curves, normalized by the integral of the experimental curve. The relative error with respect to the experimental data is between 0.119 and 0.388 for the calculations with the traditional model, depending on the temperature case. These are more than 4 times higher than the relative error achieved by the proposed model, which are in the range from 0.045 to 0.084.

The under-prediction of the moisture reduction rate when the shrinkage is not included has already been described recently in the reference [4] for drying shiitake mushroom, where a multiphase model was applied.

The computed shapes of the fruit at different times are depicted in the last column of Figure 3, and in general, they are consistent with the shape evolution shown by the captured images. This shrinking deformation proves to be an important factor for a more accurate computation of the heat a mass diffusion processes in the fruit. Therefore, the traditional model is unable to reproduce properly the moisture reduction rate in convective drying of *P. peruviana*, whereas the model including the shrinkage effects is significantly more consistent with the experimental data in the range of temperatures examined.

The Figure 5a presents the mean variation of the computed effective diffusion coefficients (*D_eff_*) as a function of the moisture ratio for the four inlet temperatures studied. The values were obtained with the Arrhenius-like relation, with the factor determined by the procedure explained in Section 3.2.2. These coefficients were used in the numerical simulations for both type of models tested, with volume change and without it. The effective diffusion coefficient varies between 1.0 × 10^−12^ m^2^ s^−1^ and 1.75 × 10^−9^ m^2^ s^−1^, with the lowest value corresponding to moisture ratios over 0.8. The diffusion coefficient increases as the moisture content is reduced, reaching a maximum at around ${\bar{C}}_{w}/{\bar{C}}_{w,initial}$ = 0.31, 0.28, 0.24 and 0.23 for 348 K, 358 K, 338 K and 328 K, respectively. Towards the end of the drying process, when the moisture content in the fruit is low, the trend is reversed and the diffusion coefficient starts decreasing, which can be attributed to the low water content of the fruit producing an exhaustion of the mass transport mechanism.

The effective diffusion coefficients increased when the drying temperature *T_in_* increased. The computed values of *D_eff_* are comparable to those reported by Vega-Gálvez et al., 2012, which ranged between 4.67 × 10^−10^ m^2^ s^−1^ and 14.9 × 10^−10^ m^2^ s^−1^ for drying of *P. peruviana* at 333–363 K. Other authors have also reported effective diffusion coefficients for berries, which are comparable to those found in this work; *D_eff_* values from 5.683 × 10^−10^ m^2^ s^−1^ to 1.555 × 10^−10^ m^2^ s^−1^ were reported for sweet cherry [39] and *D_eff_* values from 4.95 × 10^−10^ m^2^ s^−1^ to 1.42 × 10^−9^ m^2^ s^−1^ for strawberry [37].

In Figure 5b we show some 3D streamlines of the mean air flow in the dryer, with the color scale representing the magnitude of the *u*-component of the mean velocity. Note that these statistical means are the only kinematical variables explicitly resolvable by a Reynolds-averaged Navier-Stokes equation (RANS) computation. Thus, even though the computation is unsteady, the snapshot shown in this figure represents only the instantaneous mean flow, since the turbulent fluctuations have been lumped into the Reynolds stresses and its effects have been introduced through the turbulence model. The highest speeds occur at the entrance and exit of the dryer and around the fruit. Large vortices are produced inside the dryer that recirculate the air from the region near the outlet wall towards the inlet. This vortical recirculation is induced primarily by the impingement of the main flow onto the outlet wall, and secondarily by the impingement on the fruit. Some of these general flow characteristics have been observed in previous simulations carried out in a similarly shaped dryer, but with twice the size [8]. The flow pattern details are, however, specific for each case. In this case, the air in the neighborhood of the fruit tends to flow around it instead of being diverted laterally by the dried product, as in that previous research. This can be associated with the different shape and volume of product, and the different inlet velocities. An important point derived from this analysis is that the presence of this large-vortices flow pattern can produce a reduction in the drying efficiency of the equipment due to the recirculation of humid air towards the load.

Figure 6 shows lateral views (*x*-*y* plane at *z* = 0.125 m) and frontal views (*y*-*z* plane at *x* = 0.125 m) for the resulting distribution of moisture content and temperature inside and around the fruit at different time intervals. The time intervals shown in the figure have been selected with the aim of favoring better visualization of temperature and moisture profiles in the first stages of the drying process. For higher inlet temperature of the air, the gradients of moisture content and the temperature inside the fruit take higher values. The lateral and frontal views in Figure 6a show that the highest moisture contents are concentrated in the center of the fruit. At the edges, the moisture content decreases rapidly. The lateral view allows observing the shrinkage of the fruit and its effect on the moisture distribution. In this view, it is also possible to observe in all the cases a slight asymmetry in the moisture distribution near the surface between the side of the fruit that is in contact with the air jet entering the dryer and the opposite side in contact with the wake flow. Over time, drying efficiency decreases due to the reduction in the moisture concentration gradient between the fruit and the air in the entire front side of the fruit.

Figure 6b shows the isotherms in the initial drying period. At that instant, it is possible to observe in the front view how the interior temperature of the fruit rapidly increased from the surface to the center. The air temperature around the fruit hardly varied compared with the respective air inlet temperature in each case. During the initial drying period, the isotherms in all the cases and in both views have a qualitative behavior like the one observed after two minutes for *T_in_* = 358 K. The lateral view allows observing the air jet at the entrance, showing a temperature increment of the exposed surface of the fruit, with the isotherms moving in layers of decreasing values when advancing along the *x*-axis towards the wake region of the flow. In the frontal views, the isotherms exhibit a discrete radial symmetry, which is due to the symmetrical splitting of the jet when it impinges on the fruit. As the drying proceeded, the shape of the isotherms changed in lateral views mainly due to the shrinkage of the fruit.

### 3.3. Energy Consumption Results

The lowest energy consumption is obtained at 358 K (85 °C), with *E_T_* = 750.83 kJ kg^−1^, for which the necessary drying time for reaching a final moisture content of 2.01% was about 250 min. The highest energy consumption was 1663.83 kJ kg^−1^ at 338 K (65 °C), with a final moisture of 10.06%, and drying time of 780 min. The corresponding energy consumption, drying time, and final moisture for air temperatures of 328 K and 348 K, were 1488.51 kJ kg^−1^ and 1213.53 kJ kg^−1^, 900 min and 470 min, and 14.08% and 8.05%, respectively. These results show that the drying time has a high incidence on the energy consumption requirements. On the other hand, increasing the air temperature to 358 K reduced the drying time and the energy consumption, improving the drying process efficiency.

There are various reports by other authors on similar specific energy requirements, namely, for cherry [14], berberis fruit [15] and mushroom slices [16].

From the total energy consumed, the energy used to vaporize moisture in each drying process was determined. The assessment of *η_E_* shows also that the highest efficiency occurs for air at 358 K, for which *η_E_* = 68.09%, and the lowest efficiency occurs at 338 K, with *η_E_* = 30.67%. Results in-between are found for air at 328 K and 348 K, being *η_E_* = 34.22%, and *η_E_* = 41.88% respectively. Several authors working with food dried by different methods and conditions have obtained energy efficiency values of 40–65% for broccoli under convective drying [41] and 17–54% for apple slices during microwave drying [42].

## 4. Conclusions

The effect of air temperature on the drying kinetics of *P. peruviana* fruit was successfully simulated with a mathematical model solved by a 3D computational procedure. This numerical scheme includes the modeling of the asymmetric shrinkage of this fruit, which was formulated based on information acquired experimentally. The particular mode of anisotropic shrinkage of this material was determined by an image processing procedure. The numerical simulations that include this effect, through the shrinkage model introduced in this work, were compared with the traditional method of constant-volume simulation. Finally, the energy consumption for the drying of *P. peruviana* was calculated. We can make some concluding remarks:

Convective drying with air at different temperatures, in the range between 328 K (55 °C) and 358 K (85 °C), shows that the volume changes of *P. peruviana* are similar. It also shows that the volume loss is highly anisotropic, with the shrinkage occurring essentially in a direction aligned with the minor semi-axis of the ellipsoidal fruit. This anisotropic behavior can be attributable to the particular structural characteristics of the fruit studied, and it is a factor that must be taken into account when studying the convective drying of fruits.In the drying process, shrinkage can occur in one or more preferential directions and not symmetrically as is considered in most of the published research. The image capture procedure proposed in this work allowed the incorporation of experimental data into the numerical model that improved the accuracy of the simulations.A 3D computational simulation of the conjugate heat and mass transfer in a dryer and its load, which also includes the fluid dynamics of the drying air, constitutes a useful tool for analyzing the process, and provides data that would be very difficult to obtain experimentally, such as the internal temperature in the dried material.To improve the accuracy of such a kind of simulations, it is important to consider the shrinkage of the material. Furthermore, it would be advisable to assign more importance to the characteristic mode of deformation during the drying. Until now, this anisotropy has not been included in models. This work shows that the anisotropic shrinkage has a significant impact on the accuracy of a numerical simulation.More research can be done along the lines introduced in this work to characterize and calibrate models of anisotropic shrinkage of other materials.

## Figures and Tables

**Figure 1 foods-11-01880-f001:**
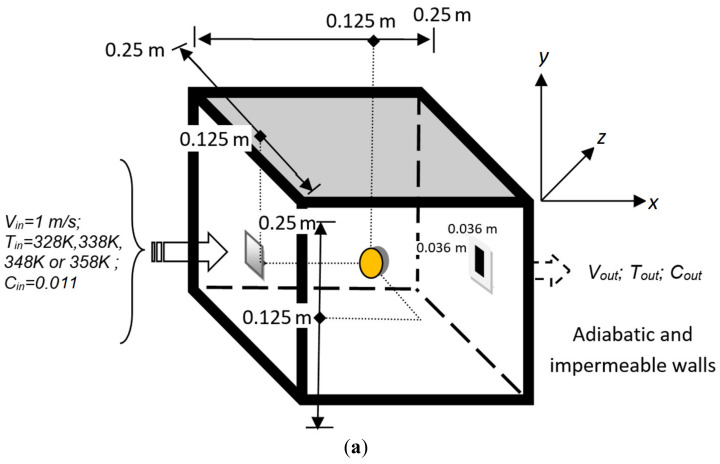
(**a**) Physical set-up for the experiments and simulations, and (**b**) optimal computational grid used in the simulations with a denser grid in the fruit zone.

**Figure 2 foods-11-01880-f002:**
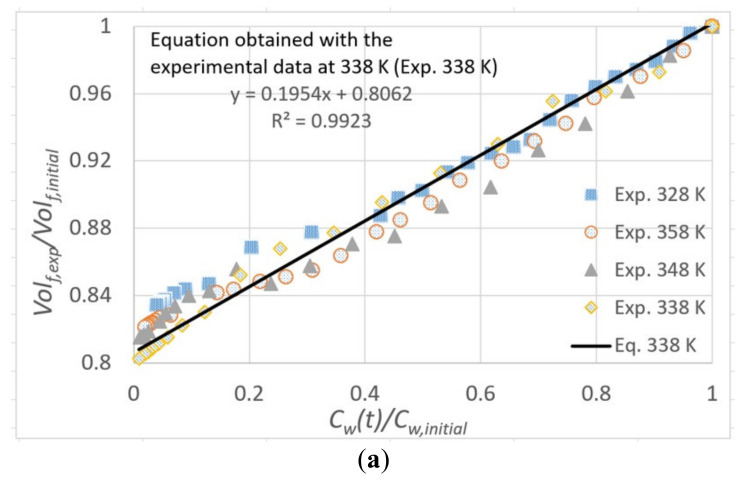
(**a**) Moisture ratio vs. volume ratio for the fruit drying. Black solid line (338 K) and equation correspond to linear equation used to calculate the volume change in the simulations. (**b**) Comparison between experimental (Exp.) and calculated drying curves at different air inlet temperatures using the traditional mathematical model without shrinkage (Calc. Tr.) and the proposed model that include shrinkage (Calc.).

**Figure 3 foods-11-01880-f003:**
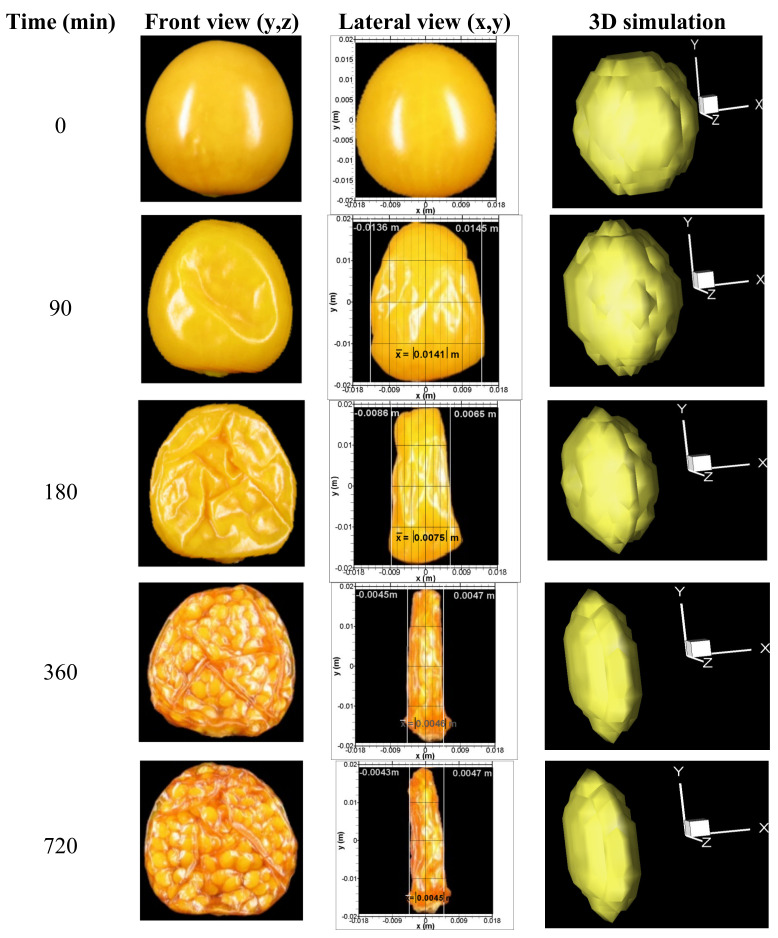
Experimental and simulated volume variation of the *P. peruviana* fruit during the drying process with air at 338 K (65 °C) inlet temperature. The reduction of the average semi-axis in the *x*-direction (*a_i_*), is measured in the lateral view.

**Figure 4 foods-11-01880-f004:**
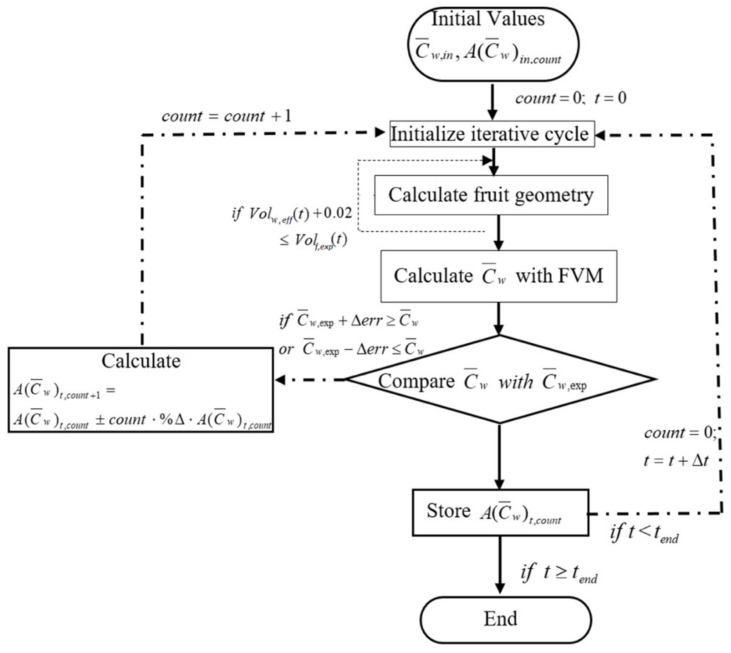
Flowchart of the algorithm used to obtain the Arrhenius factor, $A\left({\bar{C}}_{w}\right)$ and the change in volume.

**Figure 5 foods-11-01880-f005:**
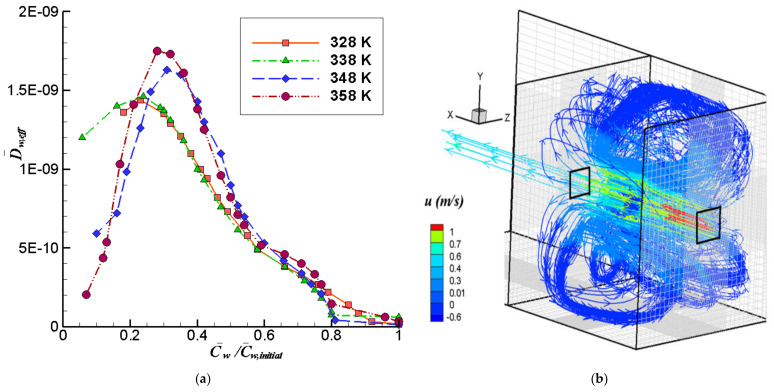
(**a**) Mean effective moisture diffusion coefficient variation as a function of dimensionless average moisture concentration. (**b**) 3D mean streamlines and u velocities (m/s) for the air flow simulation in the dryer.

**Figure 6 foods-11-01880-f006:**
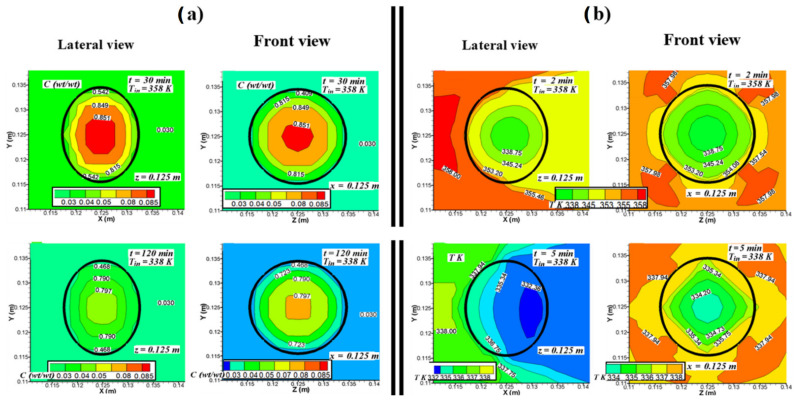
Moisture concentration and temperature distribution inside and around the *P. peruviana* fruit for lateral (*x*-*y* plane at *z* = 0.125 m) and frontal (*y*-*z* plane at *x* = 0.125 m) views at different time intervals. (**a**) Moisture after 30 min for *T_in_* = 358 K (85 °C), and after 120 min for air inlet temperature *T_in_* = 338 K (65 °C). (**b**) Temperature distribution after 2 min for *T_in_* = 358 K, and after 5 min for *T_in_* = 338 K.

**Table 1 foods-11-01880-t001:** Polynomial coefficients for the Arrhenius factor $A\left({\bar{C}}_{w}\right)$ for each drying temperature.

*T* (K)	Polynomial Coefficients						
	*a*	*b*	*c*	*d*	*e*	*f*	*g*
328	1.42 × 10^−6^	1.12 × 10^−4^	−3.88 × 10^−3^	−1.50 × 10^−3^	3.74 × 10^−3^	−3.58 × 10^−3^	1.21 × 10^−3^
338	4.20 × 10^−5^	−8.32 × 10^−4^	6.63 × 10^−3^	−2.24 × 10^−2^	3.67 × 10^−2^	−2.91 × 10^−2^	8.93 × 10^−3^
348	1.73 × 10^−5^	−4.90 × 10^−4^	5.17 × 10^−3^	−2.00 × 10^−2^	3.54 × 10^−2^	−2.98 × 10^−2^	9.63 × 10^−3^
358	1.62 × 10^−5^	−1.10 × 10^−4^	1.35 × 10^−3^	−5.85 × 10^−3^	1.11 × 10^−2^	−9.77 × 10^−3^	3.24 × 10^−3^

## Data Availability

No new data were created or analyzed in this study. Data sharing is not applicable to this article.

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
