# Peer review of "Experimental and Numerical Study of a Turbulent Air-Drying Process for an Ellipsoidal Fruit with Volume Changes"

_foods, 2022, doi:10.3390/foods11131880_

Round 1

Reviewer 1 Report

In this work the drying of fruits with three-dimensional asymmetric shrinkage was studied numerically. That's an interesting and comprehensive work. I congratulate the authors for their work. 

Author Response

Dear Reviewer

Thank you for your comments. We will check the English language and style again in this review

The authors

Reviewer 2 Report

Foods-1710230

Experimental and numerical study of a turbulent air-drying process for an ellipsoidal fruit with volume changes

It was studied a three-dimensional model for simulation, including the volume variation.

Drying was carried out with 1 m/s air at ~55, 65, 75, 85 °C.

My considerations:

Introduction: Maybe, the introduction section could be less extensive.

Material and methods:

Line 109: how much time the fruits were maintained stored?

Line 116: Please provide the temperatures also in °C.

Line 116: The air velocity was 1.0 m/s. Could the air flow be considered turbulent at this velocity? Or the consideration of turbulent flow is due to the lower air inlet and outlet areas?

Line 139: Maybe the volume determination could be better measured with liquid displacement, such as toluene.

Line 215: How was the factor A(Cw) obtained from the experimental curves of drying?

Results and discussions:

Line 288: When was determined the moisture of the fruits? After purchasing them or after the storage?

Line 291: Was this equilibrium moisture achieved in all temperatures?

Line 308: The Figure 2a is a little be confusing to understand.

Line 335: The Figure 3 shows a tendency of the fruit shrinking more in ne dimension. This fact can be occurred due to the composition of the fruit, such as vessels, and this composition can affect the moisture flow during drying, additionally to affect, this composition can be responsible for other phenomenon such as capillarity, which means that the flow could occur more in one dimension than in others. Was this behavior considered during the mathematical modelling?

Line 494: In Figure 6, it is needing a scale bar showing the moisture levels. It is confusing.

Line 512: This part of the energetic analysis must be presented in Material and Methods.

Line 518: The energy consumed should consider the difference between the temperature of drying and ambient temperature, because the dryer spends energy heating this air.

Author Response

Dear Reviewer

We thank the reviewer for his/her time reviewing our manuscript, and the comments on our work, which have improved the quality of the manuscript. In the attached document we address the concerns point by point.

Reviewer 3 Report

The work foods-1710230 deals with the development of a three-dimensional model for the simulation of Physalis peruviana drying that includes the volume changes and the asymmetry of the shrinkage. I must point out that the work is interesting and seems to present novel results regarding the analysis of non-uniform shrinkage during the drying of foods, particularly cape gooseberries.

Despite the interesting nature of the manuscript, it is presented in a rather extensive manner. I think that many of the methods could be simplified. The presentation of the results is the detail that the authors should work on the most, since the Figures need editing and better presentation. In the case of the Tables, the significant figures should be evaluated.

Here are some other recommendations:

L282, 289 and 301: P. peruviana. Correct throughout the text.

Figure 2: Improve the visual quality of Figure 2. In addition, I recommend improving spelling within the Figure, respecting the use of capital letters.

Author Response

We thank the reviewer for his/her time reviewing our manuscript, and the comments on our work, which have improved the quality of the manuscript. In the attached document we address the concerns point by point.

Reviewer 4 Report

Attached as the PDF file.

Author Response

We thank the reviewer for his/her time reviewing our manuscript, and the comments on our work, which have improved the quality of the manuscript. In the following we address the concerns point by point.

Round 2

Reviewer 3 Report

The changes have been made, so the manuscript is ready for publication.